# Evidence for a Novel Gammaherpesvirus as the Putative Agent of Malignant Catarrhal Fever Disease in Roan Antelopes (*Hippotragus equinus*)

**DOI:** 10.3390/v15030649

**Published:** 2023-02-28

**Authors:** Franziska Karola Kaiser, Madeleine de le Roi, Monica Mirolo, Sonja Tatjana Jesse, Christina Puff, Julia Bohner, Martin Ludlow, Wolfgang Baumgärtner, Albert Osterhaus

**Affiliations:** 1Research Center for Emerging Infections and Zoonoses, University of Veterinary Medicine Hannover, Foundation, Bünteweg 17, 30559 Hannover, Germany; 2Department of Pathology, University of Veterinary Medicine Hannover, Foundation, Bünteweg 17, 30559 Hannover, Germany; 3Leibniz Institute for Zoo and Wildlife Research, Alfred Kowalke Straße 17, 10315 Berlin, Germany

**Keywords:** malignant catarrhal fever, *Macavirus*, virus discovery

## Abstract

Upon the sudden death of two captive roan antelopes (*Hippotragus equinus*) that had suffered from clinical signs reminiscent of malignant catarrhal fever (MCF) in a German zoo, next generation sequencing of organ samples provided evidence of the presence of a novel gammaherpesvirus species. It shares 82.40% nucleotide identity with its so far closest relative *Alcelaphine herpesvirus 1* (AlHV-1) at the polymerase gene level. The main histopathological finding consisted of lympho-histiocytic vasculitis of the pituitary rete mirabile. The MCF-like clinical presentation and pathology, combined with the detection of a nucleotide sequence related to that of AlHV-1, indicates a spillover event of a novel member of the genus *Macavirus* of the Gammaherpesvirinae, probably from a contact species within the zoo. We propose the name *Alcelaphine herpesvirus 3* (AlHV-3) for this newly identified virus.

## 1. Introduction

Malignant catarrhal fever (MCF) is an inevitably fatal generalized lymphoproliferative disease, most frequently seen in ruminant livestock after infection with a gammaherpesvirus from another ruminant host species. These gammaherpesviruses are well adapted to their natural hosts in which they usually cause a subclinical or asymptomatic persistent infection. Interspecies transmission from their natural reservoir host into other susceptible ungulate species may result in fatal disease, commonly known as malignant catarrhal fever (MCF). Although several closely related ungulate gammaherpesviruses have been identified as etiological agents of MCF [1], this disease is observed in cattle predominantly in either of two forms, associated with the two best characterized members of the genus *Macavirus* of the Gammaherpesvirinae within the family Herpesviridae, respectively [2]. The first, sheep-associated MCF (SA-MCF), is caused by ovine herpesvirus 2 (OvHV-2) and occurs in cattle after contact with inapparently infected sheep [3]. The second, wildebeest-associated MCF (WA-MCF), is seen sporadically in African livestock after infection with *Alcelaphine herpesvirus 1* (AlHV-1), which is adapted to wildebeest (*Connochaetes taurinus*) [4]. Not only cattle are at risk of contracting MCF, as the clinical disease is more widely observed among ungulates of the family Artiodactyla that are infected with gammaherpesviruses from the same order, including deer [5], buffalos [6], antelopes [7], and pigs [8].

Clinical signs of MCF can vary depending on the species and the virus involved, but are usually dominated by progressive lethargy, salivation, nasal and ocular discharge, oral ulcerations, and corneal opacity [9,10]. Histological manifestations can be present in all affected organs and are dominated by a generalized lymphoproliferation, lymphocyctic vasculitis, and necrosis in perivascular areas, all indicative for an immune dysregulation dominated pathogenesis [11,12,13]. 

Understanding epidemiological parameters, such as natural reservoirs, susceptibility of contact species, and transmission routes both in natural habitats and in captivity, is not only key for preventing economic losses in livestock but also for protecting endangered species [14]. Especially in zoos and zoological collections, close proximity of species from different geographical origins facilitates interspecies transmission, and may pose a threat to whole herds [15]. 

## 2. Materials and Methods

### 2.1. Histopathology and Immunohistochemistry

In June 2020, a disease of unknown etiology affected a herd of captive roan antelopes (*Hippotragus equinus*) from a zoo in Lower Saxony, Germany. At necropsy, tissue samples were collected and prepared for a hematoxylin and eosin (HE) staining and light microscopy (Appendix A). In addition, immunohistochemistry for the phenotypical characterization of inflammatory cells was performed on formalin-fixed and paraffin-embedded (FFPE) tissue of the rete mirabile surrounding the pituitary gland, termed pituitary rete mirabile. Therefore, the following antibodies were used: an anti-CD3 antibody to detect T lymphocytes, an anti-CD20 antibody for the visualization of B lymphocytes, an anti-CD204 antibody, as well as an anti-Iba1 antibody for the detection of macrophages (Appendix A). Immunohistochemical investigation of the pituitary rete mirabile was analyzed semi-quantitatively (Appendix A).

### 2.2. Fluorescence in Situ Hybridization (FISH)

FISH for detection of gammaherpesvirus-specific nucleic acids was performed using the ViewRNATM ISH Tissue Core Kit (Invitrogen by Thermo Fisher Scientific, Vienna, Austria) as previously described and with minor variations [16,17]. The probe was designed accordingly to the partial sequence of 1179 base pairs generated in this study (GenBank Accession number OP113930) and commercially produced according to the manufacturer´s protocol (ViewRNATM Type 1 probe set, Life Technologies GmbH, Darmstadt). FISH for the detection of viral nucleic acids was performed on rete mirabile, cerebellum, brain stem, liver, and spleen of animal 1 and the rete mirabile of animal 2. Furthermore, formalin-fixed and paraffin-embedded (FFPE) sections of the central nervous system of a domesticated ungulate that tested positive for OvHV-2 by PCR were included. An accurate experimental procedure was conducted by testing a probe binding to an unrelated gene in a different species. Samples without probe application, but with the application of Probe Set Diluent, and roan antelopes tested negative for gammaherpesvirus by PCR were used as negative controls. Sections were evaluated using a fluorescence microscope (Olympus IX70-S8F2).

### 2.3. Virus Detection and Sequencing

Tissue samples from kidney and liver material were homogenized, centrifuged for 5 min at 7000× *g*, and RNA extraction was performed with TRIzolTM Reagent (Thermo Fisher Scientific, Waltham, MA, USA). In a first screening approach, the RNA was identified as herpesvirus-positive by applying a Herpesviridae-specific degenerative primer set [18]. For next generation sequencing, the RNA was transcribed to cDNA using a SuperScript IV Reverse Transcriptase Kit (Thermo Fisher Scientific) and with a modified sequence-independent single-primer amplification (SISPA) protocol [19] with non-ribosomal hexamers [20]. Libraries were generated by applying a Nextera XT DNA Library Preparation Kit (Illumina, San Diego, CA, USA) and sequenced on an Illumina NextSeq sequencing platform with a NextSeq 500/550 High Output kit v2.5 150 cycles (paired-end reads, 75 bp). Quality and adapter trimming, as well as assembling original reads to reference genomes, was performed using CLC Genomics Workbench 21.0 (Qiagen GmbH, Hilden, Germany). Further downstream analysis was carried out utilizing Geneious Prime (Biomatters, Ltd., Auckland, New Zealand). Primers were designed for gap filling to complete the polymerase sequence, based on the reads that matched a reference genome sequence (the primer list is available in Appendix A).

### 2.4. Evolutionary Analysis

The generated sequence was aligned to DNA polymerase catalytic subunit (ORF 9) sequences of available gammaherpesvirus genomes by multiple sequence alignment using the MAFFT software (version 7) [21]. Using MEGA X, a maximum likelihood tree with 1000 bootstraps was constructed, applying a General Time Reversible (+G, +I) model, which was calculated as a best fit [22]. To compare the 174 base pairs (bp) long partial DNA polymerase sequences of Hippotragine herpesvirus 1 (HiHV 1, also from a roan antilope; GenBank accession number NC_043060.1) to corresponding genome sections of related viruses, the Kimura 2-parameter (+I) model was utilized.

## 3. Results

### 3.1. Epidemiology and Clinical Signs

At the onset of clinical signs, the roan antelopes were kept in the same enclosure and had been in direct contact with representatives of the Artiodactyla species *Oryx leucoryx*, *Tragelaphus angasi*, *Kobus leche kafuensis*, and *Oryx dammah*. Simultaneously, the zoo was housing other antelopes, including wildebeest (*Connochaetes taurinus*) and blesbok (*Damaliscus pygargus phillipsi*) in a neighboring enclosure. The roan antelope 1 (internal identification number: S418/20) presented diarrhea, lameness of the right hind limb, anorexia, and showed a poor general condition. A few days prior to euthanasia, the second animal (internal identification number: S458/20) showed signs of poor general health, unsteadiness in walking, an increased bleeding tendency after drug applications, as well as mucosal hemorrhages.

### 3.2. Post-Mortem Examination–Pathology

Macroscopic examination of the first animal (S418/20) revealed a poor body condition, abdominal effusion, moderate to severe diffuse enlargement of lymph nodes, and petechial hemorrhages of the mucosa of the alimentary tract and of the urinary bladder. In addition, endo- and epicard displayed multifocal hemorrhages. At necropsy, the second animal (S458/20) showed pericardial and abdominal effusion and moderate enlargement of a cranial sternal lymph node and a popliteal lymph node. The skin of the dorsal part of the neck exhibited a focal alopecia with crusts. Furthermore, multifocal, subcutaneous, petechial hemorrhages were observed in the subcutis of scapula, face, and thorax.

### 3.3. Histopathology

Main histopathological findings comprised lympho-histiocytic and plasmacellular inflammation in various organs, and were located predominantly in perivascular areas but also within vascular walls. In detail, inflammatory lesions of animal 1 (S418/20) included a mild lympho-histiocytic inflammation of the pituitary rete mirabile, a mild lympho-plasmacellular vasculitis of the lung, and a mild lympho-plasmacellular periarteritis and vasculitis of the pancreas. Liver and kidney lesions were characterized by a multifocal, lympho-histiocytic, partially plasmacellular inflammation. Furthermore, the mucosa of the abomasum showed a focal necrosis. Inflammatory changes of animal 2 (S458/20) included a mild to moderate multifocal lympho-histiocytic inflammation of the pituitary rete mirabile (Figure 1), a moderate to severe multifocal lympho-histiocytic nephritis, a mild to moderate multifocal, predominantly periportal, lympho-histiocytic and plasmacellular hepatitis, as well as a multifocal, predominantly perivascular, lympho-histiocytic and partially neutrophilic dermatitis of the dorsal part of the neck.

### 3.4. Immunohistochemical Characterization and Fluorescence in Situ Hybridization (FISH) 

Perivascular and mural inflammatory cells of the pituitary rete mirabile of animal 1 (S418/20) were composed of scant to low numbers of CD204- and Iba1-positive macrophages as well as few CD3- and CD20-positive T and B lymphocytes, respectively. Immunopositive cells of animal 1 were found predominantly perivascularly. Immunohistochemical investigation of the pituitary rete mirabile of animal 2 (S458/20) revealed moderate numbers of CD3-positive T lymphocytes, predominantly perivascularly located (Figure 2A, asterisk). In addition, some vessels also exhibited CD3-positive cells within the vascular wall (Figure 2A, arrows). Furthermore, moderate numbers of CD204- and Iba1-positive macrophages were observed perivascularly as well as within the vascular wall (Figure 2B, arrows). In addition, few CD20-positive B lymphocytes were observed adjacent to vessels.

Detection of virus-specific nucleic acids by the use of FISH remained negative in antelopes as well as in a domestic ungulate that tested positive for OvHV-2 by PCR. However, a positive signal was detected by the use of a probe detecting an unrelated gene in a different species.

### 3.5. Sequencing Results and Phylogenic Comparison with Other Gammaherpesviruses

In the initial *Herpesviridae*-specific PCR a short sequence of an unknown *Macavirus* was detected. The following metagenomic sequencing revealed 156 reads that were assembled to the AlHV-1 reference genome (NC_002531.1). Based on these reads, primers could be designed to fill the genomic gaps between the individual reads. With this technique, the complete 3kb long polymerase sequence could be revealed by Sanger sequencing.

To gain further knowledge about the taxonomy of this apparent viral pathogen, the relationship of the sequence identified to those of known gammaherpesviruses was analyzed phylogenetically. Comparison of the sequence generated in this study (GenBank Accession number OP113930) with corresponding gammaherpesvirus genome sequences provides evidence for a new virus in the *Macavirus* genus. Based on this comparison, it shows closest phylogenetic relationship to viruses that have *Alcelaphinae* species as their natural reservoirs (Figure 3). Viruses with the highest nucleotide similarity in their polymerase genes are AlHV-1 (with 82.40%) and AlHV-2 (with 77.53%). For this particular gene, AlHV-1 and AlHV-2 share 82.73% of their nucleotides.

For some antelope gammaherpesviruses, only a 174 bp section of the DNA polymerase gene is currently published. This includes an entity, designated *Hippotragine herpesvirus 1* (HiHV 1; GenBank accession number NC_043060.1), which was isolated in 1991 from another roan antelope without MCF-like signs [23] and had been sequenced in 2005 [1]. Analysis based on this gene section displays an 85.06% nucleotide identity between HiHV and AlHV-1, and 83.91% of HiHV with the viral sequence identified in this study (Figure 4, Table 1).

## 4. Discussion

When representatives of the genus *Macavirus* cross the species barrier from their natural host into another ungulate species, they may cause lethal MCF. However, the identification of the natural reservoir and range of potential dead-end hosts remains challenging for many of these viruses. In the present study, we provided evidence for a novel *Macavirus*, evolutionarily most closely related to *Alcelaphine gammaherpesvirus 1*, in two captive roan antelopes that had succumbed with clinical signs and pathological lesions indicative for MCF.

Sequence analysis of the DNA polymerase gene showed 82.40% nucleotide identity between this novel viral agent and the closest relative, AlHV-1. Due to the divergence in the sequenced gene, which is comparable to the distance between AlHV-1 and AlHV-2 (81.03%), we suggest the putative name *Alcelaphine herpesvirus 3* (AlHV-3) for this apparently novel virus species. 

A hint about the epidemiological origin and natural reservoir of this newly identified virus may be obtained by focusing on the apparently related phylogenetic taxa. The two evolutionarily closest known viruses are AlHV-1 and AlHV-2, which are found in antelopes of the subfamily Alcelaphinae. AlHV-1 is frequently found in blue wildebeest (*Connochaetes taurinus*) and AlHV-2 is found in the common tsessebe (*Damaliscus lunatus*) and hartebeest (*Alcelaphus buselaphus*) [24]. Among the animals housed in the same zoo, there were two members of the subfamily Alcelaphinae, namely blue wildebeest (*Connochaetes taurinus*) and blesbok (*Damaliscus pygargus phillipsi*), which were kept in the zoo at the same time and may thus be suspected as the reservoir host. However, formally we have not proven whether this virus is indeed the causative agent of the MCF-like disease or just a coincidental detection of a well-adapted virus present in these animals. In 1991, another gammaherpesvirus (HiHV) had already been isolated from roan antelopes without MCF-related signs [23]. The presence of that virus in a roan antelope without clinical disease indicates that HiHV is more adapted to roan antelopes as its natural host than putative AlHV-3. Our analysis has demonstrated a closer relationship to AlHV-1 than to HiHV, making it less likely for the newly discovered virus to be adapted to roan antelopes. The higher nucleotide similarity to gammaherpesviruses identified in members of the subfamily Alcelaphinae than those of Hippotraginae supports the hypothesis that the novel virus in the deceased roan antelopes had spilled over from a subclinically infected member of the Alcelaphinae species, present in the zoo. We therefore speculate that one of these contact species in the same zoo was the natural reservoir for putative AlHV-3. 

Most of the histopathological lesions resemble those described for an infection with OvHV-2 or AlHV-1 causing MCF in domestic cattle, including vasculitis in various organs, particularly the pituitary rete mirabile [11,25]. It has to be considered that clinical signs and histopathological lesions can vary, depending on the affected species and the causative MCF virus. The inflammatory lesions, such as the vasculitis in the lung and pancreas of animal 1 and in the pituitary rete mirabile of animal 2, are therefore most likely associated with the putative AlHV-3 infection. Since little is known about this newly identified virus, it remains to be determined whether the additional inflammatory changes are caused or triggered by the putative AlHV-3 infection.

Immunohistochemical analysis of the pituitary rete mirabile revealed that inflammatory cells infiltrating the perivascular space as well as the vascular wall were mainly composed of moderate numbers of CD3-positive T lymphocytes and macrophages. Similar findings have been observed in MCF cases in previous studies in other species [26]. MCF is considered to represent a disease associated with the lymphoproliferation and dysregulation of T lymphocytes. The predominance of an infiltration with T lymphocytes in the presented cases is therefore concordant with findings in MCF [27].

In the present study, virus-specific nucleic acids could not be detected using FISH. In a previous study, herpesvirus-specific nucleic acids were observed by the use of in situ hybridization (ISH), but positive results could be demonstrated only in single infected cells or were limited to a few organs [28]. Similar discrepancies between the detection of MCF-associated herpesviruses by PCR and ISH have already been described by others [29]. Additionally, in contrast to ISH, PCR is supposed to be more likely positive in numerous cells with low viral load compared to a few cells containing high virus loads [30]. It should be considered that the observed lesions are possibly not directly induced by the putative AlHV-3 infection. MCF is supposed to represent an immunopathological condition and therefore a type III hypersensitivity reaction could represent the cause for the lesions [13]. Cytotoxic T lymphocytes or T-suppressor cells are assumed to initiate the lesions, but fibroblasts are also considered to be involved in the pathogenesis of the vasculitis by inducing cytokines and therefore contributing directly to the inflammatory changes seen in vessels [31]. Since the pathogenesis of MCF is still not fully understood, the inability to detect virus-specific nucleic acids should probably be interpreted as the consequence of an immune-mediated disease, triggered by a putative AlHV-3 infection [32].

Taken together, the present study has provided evidence for a novel gammaherpesvirus in two roan antelopes that had died with clinical signs and pathological lesions indicative for MCF. Sequencing and phylogenetic analysis indicated the presence of a distinct virus species, putatively designated *Alcelaphine herpesvirus 3*. Even though the original viral reservoir and the transmission route cannot be reconstructed, it is likely that this virus was the causative or triggering agent of the MCF-like disease in the roan antelopes investigated.

## Figures and Tables

**Figure 1 viruses-15-00649-f001:**
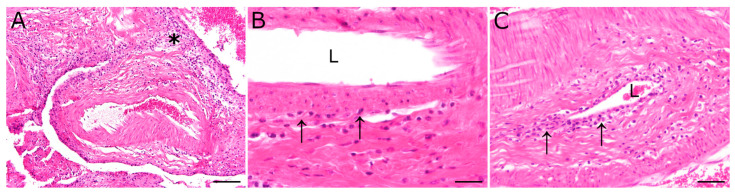
Inflammatory lesions of the pituitary rete mirabile, most prominent in animal 2, were characterized by a perivascular lympho-histiocytic inflammation (asterisk). Occasionally, vasculitis characterized by few (**B**) to moderate numbers (**C**) of mononuclear inflammatory cells (arrows), resembling lymphocytes and macrophages, were found in the vascular wall. L: vascular lumen; hematoxylin and eosin stain; scale bar (**A**): 100 µm; (**B**): 20 µm; (**C**): 50 µm.

**Figure 2 viruses-15-00649-f002:**
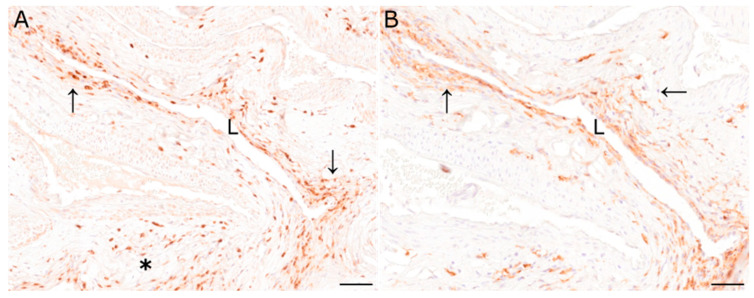
(**A**) Immunohistochemical characterization of inflammatory cells of the pituitary rete mirabile of animal 2 revealed moderate numbers of CD3-immunopositive T lymphocytes perivascularly (asterisk), and in the vascular wall (arrows). (**B**) Similarly, Iba1-positive macrophages were found in the perivascular area and the vascular wall (arrows). L: vascular lumen; immunohistochemistry; scale bar: 50 µm.

**Figure 3 viruses-15-00649-f003:**
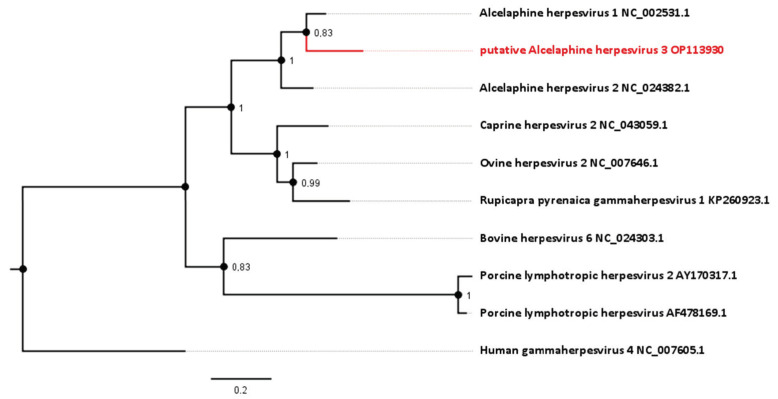
Phylogenetic analysis based on the DNA polymerase gene (ORF 9, homologue to NP_065512.1). The novel sequence from the roan antelopes, tentatively designated AlHV-3, is compared to related viruses from the genus *Macavirus*. Numbers next to branches symbolize the percentage of trees with identical taxa clusters. The scale bar indicates the number of substitutions per site.

**Figure 4 viruses-15-00649-f004:**
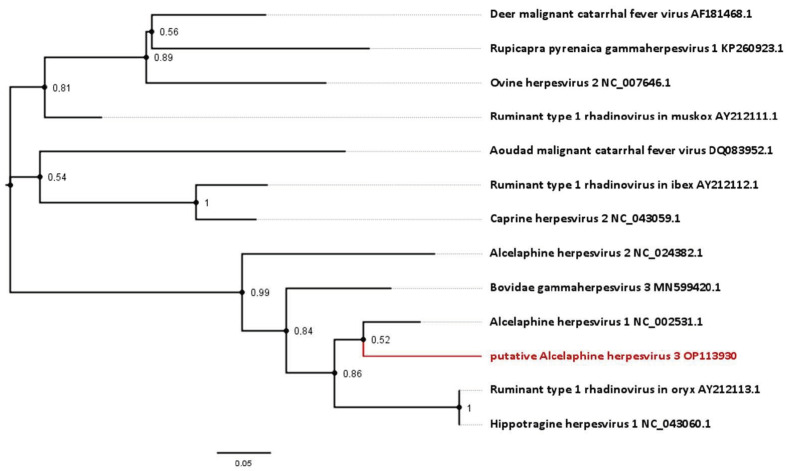
Phylogenetic analysis based on short approximately 174 bp partial sequences of the DNA polymerase available for gammaherpesviruses from antelope species. Sequences include the novel virus identified in this study, as well as the *Hippotragine herpesvirus 1*. Numbers next to branches symbolize the percentage of trees with identical taxa clusters. The scale bar indicates the number of substitutions per site.

**Table 1 viruses-15-00649-t001:** Analysis of nucleotide identity based on short 174 bp partial sequences of the DNA polymerase available for antelope gammaherpesviruses. Sequences include the novel virus identified in this study, as well as the *Hippotragine herpesvirus 1*.

	Novel Alcelaphine Herpesvirus 3 OP113930	Alcelaphine Herpesvirus 1 NC_002531.1	Hippotragine herpesvirus 1 NC_043060.1	Alcelaphine Herpesvirus 2 NC_024382.1
Novel Alcelaphine Herpesvirus 3 OP113930	100%	86.78%	83.91%	74.14%
Alcelaphine Herpesvirus 1 NC_002531.1	86.78%	100%	85.06%	81.03%
Hippotragine herpesvirus 1 NC_043060.1	83.91%	85.06%	100%	74.71%
Alcelaphine Herpesvirus 2 NC_024382.1	74.14%	81.03%	74.71%	100%

## Data Availability

The obtained sequence is publicly accessible (GenBank accession number OP113930).

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
