# Peer review of "Evidence for a Novel Gammaherpesvirus as the Putative Agent of Malignant Catarrhal Fever Disease in Roan Antelopes (Hippotragus equinus)"

_viruses, 2023, doi:10.3390/v15030649_

Round 1

Reviewer 1 Report

Congrats for authors to quality to write this manuscript. The purpose was very well structured using different methodologies to verify the putative virus.

Author Response

We thank the reviewer for his comments and the feedback.

Reviewer 2 Report

Kaiser and colleagues describe the identification of DNA sequences of a novel gammaherpesvirus after fatal infection of two captive roan antilopes that succumbed to disease clinically similar to malignant catarrhal fever. They also provide a case report with histology of these cases. This is interesting and should definitely be published, but some more information regarding the sequence(s) that were generated would be helpful.

Specific:

Which rete mirabile was analyzed, or is there only one in these animals or are there different retia mirabilia? Veterinarian jargon should be made more accessible to readers without veterinarian/medical background as this is not the typical or sole target audience of Viruses.

Lines 73-82: Read length, paired/unpaired, depth (number of reads total, number of reads mapping to the novel virus, coverage should be given.

As this is not given in the manuscript (space limitations?), there should be ample space for that in the appendix.

I am either misreading the manuscript or the result section completely lacks any mention of sequencing and its results, but suddenly a virus-specific nucleic acid probe is used in FISH and a phylogenetic comparison is made. This needs to be presented in a different manner and is highly confusing unless one thoroughly reads the methods and technical appendix beforehand.

Sequence of the FISH probe should be specified.

Line 158: There should not be a comma at the end of the line after sequences and either comparison should be put into plural or the verb into singular.

Accession OP113930 does not seem to exist. This sequence should be made available to the reviewers.

Reviewer 3 Report

This is very interesting case report. The pathological, histopathological, and immunohistochemical findings in affected roan antelopes are consistent with MCF, and the sequence of the polymerase gene suggests infection with a novel virus closely related to ALHV-1. However, the data are insufficient to assign a new virus name “Alcelaphine herpesvirus 3”, and virus isolation or full-length genome sequencing is necessary.

As the authors point out, there are many unknowns regarding animal-to-animal virus transmission. It would be better try investigate for viral infection in the Artiodactyla species that were in contact with diseased roan antelopes by virus isolation from peripheral blood or detection of antibodies to ALHV.
